# MOReL: Model-Based Offline Reinforcement Learning

**Rahul Kidambi**[*]
Cornell University, Ithaca
rkidambi@cornell.edu

**Aravind Rajeswaran**[*]
University of Washington, Seattle
Google Research, Brain Team
aravraj@cs.washington.edu

**Praneeth Netrapalli**
Microsoft Research, India
praneeth@microsoft.com

**Thorsten Joachims**
Cornell University, Ithaca
tj@cs.cornell.edu

## Abstract

In offline reinforcement learning (RL), the goal is to learn a highly rewarding policy based solely on a dataset of historical interactions with the environment. The ability to train RL policies offline would greatly expand where RL can be applied, its data efficiency, and its experimental velocity. Prior work in offline RL has been confined almost exclusively to model-free RL approaches. In this work, we present `MOReL`, an algorithmic framework for model-based offline RL. This framework consists of two steps: (a) learning a *pessimistic MDP* (P-MDP) using the offline dataset; (b) learning a near-optimal policy in this P-MDP. The learned P-MDP has the property that for *any* policy, the performance in the real environment is approximately lower-bounded by the performance in the P-MDP. This enables it to serve as a good surrogate for purposes of policy evaluation and learning, and overcome common pitfalls of model-based RL like model exploitation. Theoretically, we show that `MOReL` enjoys strong performance guarantees for offline RL. Through experiments, we show that `MOReL` matches or exceeds state-of-the-art results in widely studied offline RL benchmarks. Moreover, the modular design of `MOReL` enables future advances in its components (e.g., in model learning, planning etc.) to directly translate into improvements for offline RL. Project webpage: https://sites.google.com/view/morel

## 1 Introduction

The fields of computer vision and NLP have seen tremendous advances by utilizing large-scale offline datasets [1, 2, 3] for training and deploying deep learning models [4, 5, 6, 7]. In contrast, reinforcement learning (RL) [8] is typically viewed as an online learning process. The RL agent iteratively collects data through interactions with the environment while learning the policy. Unfortunately, a direct embodiment of this trial and error learning is often inefficient and feasible only with a simulator [9, 10, 11]. Similar to progress in other fields of AI, the ability to learn from offline datasets may hold the key to unlocking the sample efficiency and widespread use of RL agents.

Offline RL, also known as batch RL [12], involves learning a highly rewarding policy using only a static offline dataset collected by one or more data logging (behavior) policies. Since the data has already been collected, offline RL abstracts away data collection or exploration, and allows prime focus on data-driven learning of policies. This abstraction is suitable for safety sensitive applications

---

[*]Equal Contributions. Correspond to rkidambi@cornell.edu and aravraj@cs.washington.edu.

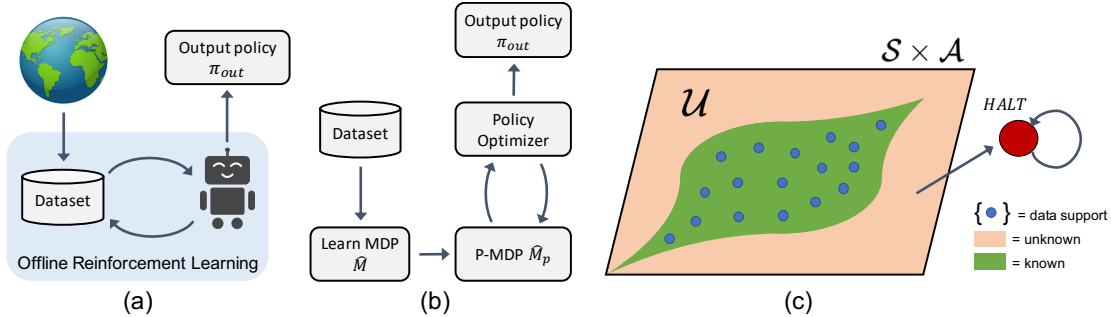

Figure 1: (a) Illustration of the offline RL paradigm. (b) Illustration of our framework, MOReL, which learns a pessimistic MDP (P-MDP) from the dataset and uses it for policy search. (c) Illustration of the P-MDP, which partitions the state-action space into known (green) and unknown (orange) regions, and also forces a transition to a low reward absorbing state (HALT) from unknown regions. Blue dots denote the support in the dataset. See algorithm 1 for more details.

like healthcare and industrial automation where careful oversight by a domain expert is necessary for taking exploratory actions or deploying new policies [13, 14]. Additionally, large historical datasets are readily available in domains like autonomous driving and recommendation systems, where offline RL may be used to improve upon currently deployed policies.

Due to use of static dataset, offline RL faces unique challenges. Over the course of learning, the agent has to evaluate and reason about various candidate policy updates. This *offline policy evaluation* is particularly challenging due to deviation between the state visitation distribution of the candidate policy and the logging policy. Furthermore, this difficulty is exacerbated over the course of learning as the candidate policies increasingly deviate from the logging policy. This change in distribution, as a result of policy updates, is typically called *distribution shift* and constitutes a major challenge in offline RL. Recent studies show that directly using off-policy RL algorithms with an offline dataset yields poor results due to distribution shift and function approximation errors [15, 16, 17]. To overcome this, prior works have proposed modifications like Q-network ensembles [15, 18] and regularization towards the data logging policy [19, 16, 18]. Most notably, prior work in offline RL has been confined almost exclusively to model-free methods [20, 15, 16, 19, 17, 18, 21].

Model-based RL (MBRL) presents an alternate set of approaches involving the learning of approximate dynamics models which can subsequently be used for policy search. MBRL enables the use of generic priors like smoothness and physics [22] for model learning, and a wide variety of planning algorithms [23, 24, 25, 26, 27]. As a result, MBRL algorithms have been highly sample efficient for online RL [28, 29]. However, direct use of MBRL algorithms with offline datasets can prove challenging, again due to the distribution shift issue. In particular, since the dataset may not span the entire state-action space, the learned model is unlikely to be globally accurate. As a result, planning using a learned model without any safeguards against model inaccuracy can result in *"model exploitation"* [30, 31, 29, 28], yielding poor results [32]. In this context, we study the pertinent question of how to effectively regularize and adapt model-based methods for offline RL.

**Our Contributions:** The principal contribution of our work is the development of MOReL (Model-based Offline Reinforcement Learning), a novel model-based framework for offline RL (see figure 1 for an overview). MOReL enjoys rigorous theoretical guarantees, enables transparent algorithm design, and offers state of the art (SOTA) results on widely studied offline RL benchmarks.

- MOReL consists of two modular steps: (a) learning a *pessimistic MDP* (P-MDP) using the offline dataset; and (b) learning a near-optimal policy for the P-MDP. For *any* policy, the performance in the true MDP (environment) is approximately lower bounded by the performance in the P-MDP, making it a suitable surrogate for purposes of policy evaluation and learning. This also guards against model exploitation, which often plagues MBRL.

- The P-MDP partitions the state space into "known" and "unknown" regions, and uses a large negative reward for unknown regions. This provides a regularizing effect during policy learning by heavily penalizing policies that visit unknown states. Such regularization in the space of state visitations, afforded by a model-based approach, is particularly well suited for offline RL. In contrast, model-free algorithms [16, 18] are forced to regularize the policies directly towards the data logging policy, which can be overly conservative.

- Theoretically, we establish upper bounds for the sub-optimality of a policy learned with `MOReL`, and a worst case lower-bound for the sub-optimality of a policy learnable by *any* offline RL algorithm. We find that these bounds match upto log factors, suggesting that the performance guarantee of `MOReL` is nearly optimal in terms of discount factor and support mismatch between optimal and data collecting policies (see Corollary 3 and Proposition 4).
- We evaluate `MOReL` on standard benchmark tasks used for offline RL. `MOReL` obtains SOTA results in 12 out of 20 environment-dataset configurations, and performs competitively in the rest. In contrast, the best prior algorithm [18] obtains SOTA results in only 5 (out of 20) configurations.

## 2 Related Work

Offline RL dates to at least the work of Lange et al. [12], and has applications in healthcare [33, 34, 35], recommendation systems [36, 37, 38, 39], dialogue systems [40, 19, 41], and autonomous driving [42]. Algorithms for offline RL typically fall under three categories. The first approach utilizes **importance sampling** and is popular in contextual bandits [43, 36, 37]. For full offline RL, Liu et al. [44] perform planning with learned importance weights [45, 46, 47] while using a notion of pessimism for regularization. However, Liu et al. [44] don't explicitly consider generalization and their guarantees become degenerate if the logging policy does not span the support of the optimal policy. In contrast, our approach accounts for generalization, leads to stronger theoretical guarantees, and obtains SOTA results on challenging offline RL benchmarks. The second, and perhaps most popular approach is based on **approximate dynamic programming (ADP)**. Recent works have proposed modification to standard ADP algorithms [48, 49, 50, 51] towards stabilizing Bellman targets with ensembles [17, 15, 19] and regularizing the learned policy towards the data logging policy [15, 16, 18]. ADP-based offline RL has also be studied theoretically [26, 52]. However, these works again don't study the impact of support mismatch between logging policy and optimal policy. Finally, **model-based RL** has been explored only sparsely for offline RL in literature [32, 53] (see appendix for details). The work of Ross and Bagnell [32] considered a straightforward approach of learning a model from offline data, followed by planning. They showed that this can have arbitrarily large sub-optimality. In contrast, our work develops a new framework utilizing the notion of pessimism, and shows both theoretically and experimentally that MBRL can be highly effective for offline RL. Concurrent to our work, Yu et al. [54] also study a model-based approach to offline RL.

A cornerstone of `MOReL` is the P-MDP which partitions the state space into known and unknown regions. Such a hard partitioning was considered in early works like $E^3$ [55], R-MAX [56], and metric-$E^3$ [57], but was not used to encourage pessimism. Similar ideas have been explored in related settings like online RL [58, 59] and imitation learning [60]. Our work differs in its focus on offline RL, where we show the P-MDP construction plays a crucial role. Moreover, direct practical instantiations of $E^3$ and metric-$E^3$ with function approximation have remained elusive.

## 3 Problem Formulation

A **Markov Decision Process (MDP)** is represented by $\mathcal{M} = \{S, A, r, P, \rho_0, \gamma\}$, where, $S$ is the state-space, $A$ is the action-space, $r : S \times A \rightarrow [-R_{\max}, R_{\max}]$ is the reward function, $P : S \times A \times S \rightarrow \mathbb{R}_+$ is the transition kernel, $\rho_0$ is the initial state distribution, and $\gamma$ the discount factor. A policy defines a mapping from states to a probability distribution over actions, $\pi : S \times A \rightarrow \mathbb{R}_+$. The goal is to obtain a policy that maximizes expected performance with states sampled according to $\rho_0$, i.e.:

$$\max_{\pi} \ J_{\rho_0}(\pi, \mathcal{M}) := \mathbb{E}_{s \sim \rho_0} \left[ V^{\pi}(s, \mathcal{M}) \right], \text{ where, } V^{\pi}(s, \mathcal{M}) = \mathbb{E} \left[ \sum_{t=0}^{\infty} \gamma^t r(s_t, a_t) | s_0 = s \right]. \quad (1)$$

To avoid notation clutter, we suppress the dependence on $\rho_0$ when understood from context, i.e. $J(\pi, \mathcal{M}) \equiv J_{\rho_0}(\pi, \mathcal{M})$. We denote the optimal policy using $\pi^* := \arg\max_{\pi} J_{\rho_0}(\pi, \mathcal{M})$. Typically, a class of parameterized policies $\pi_{\theta} \in \Pi(\Theta)$ are considered, and the parameters $\theta$ are optimized.

In **offline RL**, we are provided with a static dataset of interactions with the environment consisting of $\mathcal{D} = \{(s_i, a_i, r_i, s_i')\}_{i=1}^{N}$. The data can be collected using one or more logging (or behavioral) policies denoted by $\pi_b$. We do not assume logging policies are known in our formulation. Given $\mathcal{D}$, the goal in offline RL is to output a $\pi_{\text{out}}$ with minimal sub-optimality, i.e. $J(\pi^*, \mathcal{M}) - J(\pi_{\text{out}}, \mathcal{M})$. In general, it may not be possible to learn the optimal policy with a static dataset (see section 4.1). Thus, we aim to design algorithms that would result in as low sub-optimality as possible.

**Model-Based RL (MBRL)** involves learning an MDP $\hat{\mathcal{M}} = \{S, A, r, \hat{P}, \hat{\rho}_0, \gamma\}$ which uses the learned transitions $\hat{P}$ instead of the true transition dynamics $P$. In this paper, we assume the reward function $r$ is known and use it in $\hat{M}$. If $r(\cdot)$ is unknown, it can also be learned from data. The initial state distribution $\hat{\rho}_0$ can either be learned from the data or $\rho_0$ can be used if known. Analogous to $\mathcal{M}$, we use $J_{\hat{\rho}_0}(\pi, \hat{\mathcal{M}})$ or simply $J(\pi, \hat{\mathcal{M}})$ to denote performance of $\pi$ in $\hat{M}$.

## 4 Algorithmic Framework

For ease of exposition and clarity, we first begin by presenting an idealized version of MOReL, for which we also establish theoretical guarantees. Subsequently, we describe a practical version of MOReL that we use in our experiments. Algorithm 1 presents the broad framework of MOReL. We now study each component of MOReL in greater detail.

---

**Algorithm 1** MOReL: Model Based Offline Reinforcement Learning

---

1: **Require** Dataset $\mathcal{D}$
2: Learn approximate dynamics model $\hat{P} : S \times A \rightarrow S$ using $\mathcal{D}$.
3: Construct $\alpha$-USAD, $U^\alpha : S \times A \rightarrow \{\text{TRUE}, \text{FALSE}\}$ using $\mathcal{D}$. (see Definition 1).
4: Construct the *pessimistic* MDP $\hat{\mathcal{M}}_p = \{S \cup \text{HALT}, A, r_p, \hat{P}_p, \hat{\rho}_0, \gamma\}$. (see Definition 2).
5: (OPTIONAL) Use a behavior cloning approach to estimate the behavior policy $\hat{\pi}_b$.
6: $\pi_{\text{out}} \leftarrow \text{PLANNER}(\hat{\mathcal{M}}_p, \pi_{\text{init}} = \hat{\pi}_b)$
7: **Return** $\pi_{\text{out}}$.

---

**Learning the dynamics model:** The first step involves using the offline dataset to learn an approximate dynamics model $\hat{P}(\cdot|s, a)$. This can be achived through maximum likelihood estimation or other techniques from generative and dynamics modeling [61, 62, 63]. Since the offline dataset may not span the entire state space, the learned model may not be globally accurate. So, a naïve MBRL approach that directly plans with the learned model may over-estimate rewards in unfamiliar parts of the state space, resulting in a highly sub-optimal policy [32]. We overcome this with the next step.

**Unknown state-action detector (USAD):** We partition the state-action space into known and unknown regions based on the accuracy of learned model as follows.

**Definition 1.** *($\alpha$-USAD) Given a state-action pair $(s, a)$, define an unknown state action detector as:*

$$U^\alpha(s, a) = \begin{cases} \text{FALSE} \ \ (i.e.\ Known) & \text{if } D_{TV}\left(\hat{P}(\cdot|s, a), P(\cdot|s, a)\right) \leq \alpha \ \text{can be guaranteed} \\ \text{TRUE} \ \ (i.e.\ Unknown) & \text{otherwise} \end{cases} \tag{2}$$

Intuitively, USAD provides confidence about where the learned model is accurate. It flags state-actions for which the model is guarenteed to be accurate as "known", while flagging state-actions where such a guarantee cannot be ascertained as "unknown". Note that USAD is based on the ability to guarantee the accuracy, and is not an inherent property of the model. In other words, there could be states where the model is actually accurate, but flagged as unknown due to the agent's inability to guarantee accuracy. Two factors contribute to USAD's effectiveness: (a) data availability: having sufficient data points "close" to the query; (b) quality of representations: certain representations, like those based on physics, can lead to better generalization guarantees. This suggests that larger datasets and research in representation learning can potentially enable stronger offline RL results.

**Pessimistic MDP construction:** We now construct a pessimistic MDP (P-MDP) using the learned model and USAD, which penalizes policies that venture into unknown parts of state-action space.

**Definition 2.** *The $(\alpha, \kappa)$-pessimistic MDP is described by $\hat{\mathcal{M}}_p := \{S \cup HALT, A, r_p, \hat{P}_p, \hat{\rho}_0, \gamma\}$. Here, $S$ and $A$ are states and actions in the MDP $\mathcal{M}$. HALT is an additional absorbing state we introduce into the state space of $\hat{\mathcal{M}}_p$. $\hat{\rho}_0$ is the initial state distribution learned from the dataset $\mathcal{D}$. $\gamma$ is the discount factor (same as $\mathcal{M}$). The modified reward and transition dynamics are given by:*

$$\hat{P}_p(s'|s, a) = \begin{cases} \delta(s' = \text{HALT}) & \text{if } U^\alpha(s, a) = \text{TRUE} \\ & \text{or } s = \text{HALT} \\ \hat{P}(s'|s, a) & \text{otherwise} \end{cases} \qquad r_p(s, a) = \begin{cases} -\kappa & \text{if } s = \text{HALT} \\ r(s, a) & \text{otherwise} \end{cases}$$

$\delta(s' = \text{HALT})$ is the Dirac delta function, which forces the MDP to transition to the absorbing state HALT. For unknown state-action pairs, we use a reward of $-\kappa$, while all known state-actions receive the same reward as in the environment. The P-MDP heavily punishes policies that visit unknown states, thereby providing a safeguard against distribution shift and model exploitation.

**Planning:** The final step in MOReL is to perform planning in the P-MDP defined above. For simplicity, we assume a planning oracle that returns an $\epsilon_\pi$-sub-optimal policy in the P-MDP. A number of algorithms based on MPC [23, 64], search-based planning [65, 25], dynamic programming [49, 26], or policy optimization [27, 51, 66, 67] can be used to approximately realize this.

### 4.1 Theoretical Results

In order to state our results, we begin by defining the notion of hitting time.

**Definition 3.** *(Hitting time) Given an MDP $\mathcal{M}$, starting state distribution $\rho_0$, state-action pair $(s, a)$ and a policy $\pi$, the* hitting time $T^\pi_{(s,a)}$ *is defined as the random variable denoting the first time action $a$ is taken at state $s$ by $\pi$ on $\mathcal{M}$, and is equal to $\infty$ if $a$ is never taken by $\pi$ from state $s$. For a set of state-action pairs $\mathcal{S} \subseteq S \times A$, we define $T^\pi_{\mathcal{S}} \overset{def}{=} \min_{(s,a) \in \mathcal{S}} T^\pi_{(s,a)}$.*

We are now ready to present our main result with the proofs deferred to the appendix.

**Theorem 1.** *(Policy value with pessimism) The value of any policy $\pi$ on the original MDP $\mathcal{M}$ and its $(\alpha, R_{\max})$-pessimistic MDP $\hat{\mathcal{M}}_p$ satisfies:*

$$J_{\hat{\rho}_0}(\pi, \hat{\mathcal{M}}_p) \geq J_{\rho_0}(\pi, \mathcal{M}) - \frac{2R_{max}}{1-\gamma} \cdot D_{TV}(\rho_0, \hat{\rho}_0) - \frac{2\gamma R_{max}}{(1-\gamma)^2} \cdot \alpha - \frac{2R_{max}}{1-\gamma} \cdot \mathbb{E}\left[\gamma^{T^\pi_{\mathcal{U}}}\right], \text{ and}$$

$$J_{\hat{\rho}_0}(\pi, \hat{\mathcal{M}}_p) \leq J_{\rho_0}(\pi, \mathcal{M}) + \frac{2R_{max}}{1-\gamma} \cdot D_{TV}(\rho_0, \hat{\rho}_0) + \frac{2\gamma R_{max}}{(1-\gamma)^2} \cdot \alpha,$$

*where $T^\pi_{\mathcal{U}}$ denotes the hitting time of unknown states $\mathcal{U} \overset{def}{=} \{(s,a) : U^\alpha(s,a) = \text{TRUE}\}$ by $\pi$ on $\mathcal{M}$.*

Theorem 1 can be used to bound the suboptimality of output policy $\pi_{\text{out}}$ of Algorithm 1.

**Corollary 2.** *Suppose PLANNER in Algorithm 1 returns an $\epsilon_\pi$ sub-optimal policy. Then, we have*

$$J_{\rho_0}(\pi^*, \mathcal{M}) - J_{\rho_0}(\pi_{out}, \mathcal{M}) \leq \epsilon_\pi + \frac{4R_{max}}{1-\gamma} \cdot D_{TV}(\rho_0, \hat{\rho}_0) + \frac{4\gamma R_{max}}{(1-\gamma)^2} \cdot \alpha + \frac{2R_{max}}{1-\gamma} \cdot \mathbb{E}\left[\gamma^{T^{\pi^*}_{\mathcal{U}}}\right].$$

Theorem 1 indicates that the difference in any policy $\pi$'s value in the $(\alpha, R_{\max})$ pessimistic MDP $\hat{\mathcal{M}}_p$ and the original MDP $\mathcal{M}$ depends on: i) the total variation distance between the true and learned starting state distribution $D_{TV}(\rho_0, \hat{\rho}_0)$, ii) the maximum total variation distance $\alpha$ between the learned model $\hat{P}(\cdot|s,a)$ and the true model $P(\cdot|s,a)$ over all *known* states i.e., $\{(s,a)|U^\alpha(s,a) = \text{FALSE}\}$ and, iii) the hitting time $T^{\pi^*}_{\mathcal{U}}$ of unknown states $\mathcal{U}$ on the original MDP $\mathcal{M}$ under the optimal policy $\pi^*$. As the dataset size increases, $D_{TV}(\rho_0, \hat{\rho}_0)$ and $\alpha$ approach zero, indicating $\mathbb{E}\left[\gamma^{T^{\pi^*}_{\mathcal{U}}}\right]$ determines the sub-optimality in the limit. For comparison to prior work, Lemma 5 in Appendix A bounds this quantity in terms of state-action visitation distribution, which for a policy $\pi$ on $\mathcal{M}$ is expressed as $d^{\pi,\mathcal{M}}(s,a) \overset{def}{=} (1-\gamma)\sum_{t=0}^{\infty} \gamma^t P(s_t = s, a_t = a | s_0 \sim \rho_0, \pi, \mathcal{M})$. We have the following corollary:

**Corollary 3.** *(Upper bound) Suppose the dataset $\mathcal{D}$ is large enough so that $\alpha = D_{TV}(\rho_0, \hat{\rho}_0) = 0$. Then, the output $\pi_{out}$ of Algorithm 1 satisfies:*

$$J_{\rho_0}(\pi^*, \mathcal{M}) - J_{\rho_0}(\pi_{out}, \mathcal{M}) \leq \epsilon_\pi + \frac{2R_{max}}{1-\gamma} \cdot \mathbb{E}\left[\gamma^{T^{\pi^*}_{\mathcal{U}}}\right] \leq \epsilon_\pi + \frac{2R_{\max}}{(1-\gamma)^2} \cdot d^{\pi^*,\mathcal{M}}(\mathcal{U})$$

Prior results [15, 44] assume that $d^{\pi^*,\mathcal{M}}(\mathcal{U}_D) = 0$, where $\mathcal{U}_D \overset{def}{=} \{(s,a)|(s,a,r,s') \notin \mathcal{D}\} \supseteq \mathcal{U}$ is the set of state action pairs that don't occur in the offline dataset, and guarantee finding an optimal policy under this assumption. Our result significantly improves upon these in three ways: i) $\mathcal{U}_D$ is replaced by a smaller set $\mathcal{U}$, leveraging the generalization ability of learned dynamics model, ii) the sub-optimality bound is extended to the setting where full support coverage is not satisfied i.e., $d^{\pi^*,\mathcal{M}}(\mathcal{U}) > 0$, and iii) the sub-optimality bound on $\pi_{\text{out}}$ is stated in terms of unknown state hitting time $T^{\pi^*}_{\mathcal{U}}$, which can be significantly better than a bound that depends only on $d^{\pi^*,\mathcal{M}}(\mathcal{U})$. To further strengthen our results, the following proposition shows that Corollary 3 is tight up to $\log$ factors.

**Proposition 4.** *(Lower bound) For any discount factor* $\gamma \in [0.95, 1)$*, support mismatch* $\epsilon \in \left(0, \frac{1-\gamma}{\log \frac{1}{1-\gamma}}\right]$ *and reward range* $[-R_{max}, R_{max}]$*, there is an MDP* $\mathcal{M}$*, starting state distribution* $\rho_0$*, optimal policy* $\pi^*$ *and a dataset collection policy* $\pi_b$ *such that i)* $d^{\pi^*, \mathcal{M}}(\mathcal{U}_D) \leq \epsilon$*, and ii) any policy* $\hat{\pi}$ *that is learned solely using the dataset collected with* $\pi_b$ *satisfies:*

$$J_{\rho_0}(\pi^*, \mathcal{M}) - J_{\rho_0}(\hat{\pi}, \mathcal{M}) \geq \frac{R_{max}}{4(1-\gamma)^2} \cdot \frac{\epsilon}{\log \frac{1}{1-\gamma}},$$

*where* $\mathcal{U}_D \overset{def}{=} \{(s, a) : (s, a, r, s') \notin \mathcal{D} \text{ for any } r, s'\}$ *denotes state action pairs not in the dataset* $\mathcal{D}$. We see that for $\epsilon < (1-\gamma)/(\log \frac{1}{1-\gamma})$, the lower bound obtained by Proposition 4 on the suboptimality of any offline RL algorithm matches the upper bound of Corollary 3 up to an additional log factor. For $\epsilon > (1 - \gamma)/(\log \frac{1}{1-\gamma})$, Proposition 4 also implies (by choosing $\epsilon' = (1 - \gamma)/(\log \frac{1}{1-\gamma}) < \epsilon$) that any offline algorithm must suffer at least constant factor suboptimality in the worst case. Finally, we note that as the size of dataset $\mathcal{D}$ increases to $\infty$, Theorem 1 and the optimality of PLANNER together imply that $J_{\rho_0}(\pi_{\text{out}}, \mathcal{M}) \geq J_{\rho_0}(\pi_b, \mathcal{M})$ since $\mathbb{E}\left[\gamma^{T_{\mathcal{U}}^{\pi_b}}\right]$ goes to 0. The proof is similar to that of Corollary 3 and is presented in Appendix A.

### 4.2 Practical Implementation Of `MOReL`

We now present a practical instantiation of `MOReL` (algorithm 1) utilizing a recent model-based NPG approach [28]. The principal difference is the specialization to offline RL and construction of the P-MDP using an ensemble of learned dynamics models.

**Dynamics model learning:** We consider Gaussian dynamics models [28], $\hat{P}(\cdot|s, a) \equiv \mathcal{N}(f_\phi(s, a), \Sigma)$, with mean $f_\phi(s, a) = s + \sigma_\Delta \text{ MLP}_\phi((s - \mu_s)/\sigma_s, (a - \mu_a)/\sigma_a)$, where $\mu_s, \sigma_s, \mu_a, \sigma_a$ are the mean and standard deviations of states/actions in $\mathcal{D}$; $\sigma_\Delta$ is the standard deviation of state differences, i.e. $\Delta = s' - s, (s, s') \in \mathcal{D}$; this parameterization ensures local continuity since the MLP learns only the state differences. The MLP parameters are optimized using maximum likelihood estimation with mini-batch stochastic optimization using Adam [68].

**Unknown state-action detector (USAD):** In order to partition the state-action space into known and unknown regions, we use uncertainty quantification [69, 70, 71, 72]. In particular, we consider approaches that track uncertainty using the predictions of ensembles of models [69, 72]. We learn multiple models $\{f_{\phi_1}, f_{\phi_2}, \ldots\}$ where each model uses a different weight initialization and are optimized with different mini-batch sequences. Subsequently, we compute the ensemble discrepancy as $\text{disc}(s, a) = \max_{i,j} \left\| f_{\phi_i}(s, a) - f_{\phi_j}(s, a) \right\|_2$, where $f_{\phi_i}$ and $f_{\phi_j}$ are members of the ensemble. With this, we implement USAD as below, with threshold being a tunable hyperparameter.

$$U_{\text{practical}}(s, a) = \begin{cases} \text{FALSE (i.e. Known)} & \text{if } \text{disc}(s, a) \leq \text{threshold} \\ \text{TRUE (i.e. Unknown)} & \text{if } \text{disc}(s, a) > \text{threshold} \end{cases}. \tag{3}$$

## 5 Experiments

Through our experimental evaluation, we aim to answer the following questions:

1. **Comparison to prior work:** How does `MOReL` compare to prior SOTA offline RL algorithms [15, 16, 18] in commonly studied benchmark tasks?
2. **Quality of logging policy:** How does the quality (value) of the data logging (behavior) policy, and by extension the dataset, impact the quality of the policy learned by `MOReL`?
3. **Importance of pessimistic MDP:** How does `MOReL` compare against a naïve model-based RL approach that directly plans in a learned model without any safeguards?
4. **Transfer from pessimistic MDP to environment:** Does learning progress in the P-MDP, which we use for policy learning, effectively translate or transfer to learning progress in the environment?

To answer the above questions, we consider commonly studied benchmark tasks from OpenAI gym [73] simulated with MuJoCo [74]. Our experimental setup closely follows prior work [15, 16, 18]. The tasks considered include `Hopper-v2`, `HalfCheetah-v2`, `Ant-v2`, and `Walker2d-v2`, which

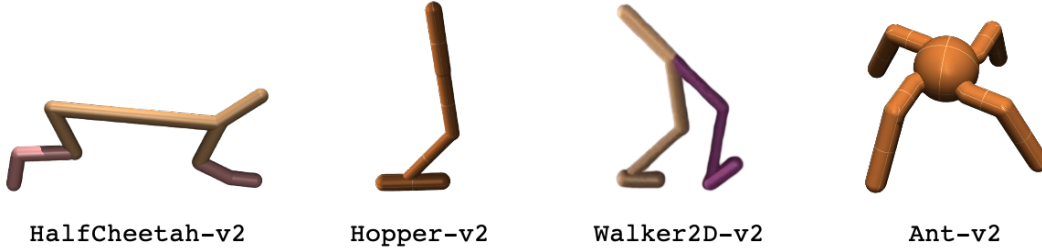

HalfCheetah-v2     Hopper-v2     Walker2D-v2     Ant-v2

Figure 2: Illustration of the suite of tasks considered in this work. These tasks require the RL agent to learn locomotion gaits for the illustrated simulated characters.

are illustrated in Figure 2. We consider five different logged data-sets for each environment, totalling 20 environment-dataset combinations. Datasets are collected based on the work of Wu et al. [18], with each dataset containing the equivalent of 1 million timesteps of environment interaction. We first partially train a policy ($\pi_p$) to obtain values around 1000, 4000, 1000, and 1000 respectively for the four environments. The first exploration strategy, `Pure`, involves collecting the dataset solely using $\pi_p$. The four other datasets are collected using a combination of $\pi_p$, a *noisy* variant of $\pi_p$, and an untrained random policy. The noisy variant of $\pi_p$ utilizes either epsilon-greedy or Gaussian noise, resulting in configurations `eps-1`, `eps-3`, `gauss-1`, `gauss-3` that signify various types and magnitudes of noise added to $\pi_p$. Please see appendix for additional experimental details.

We parameterize the dynamics model using 2-layer ReLU-MLPs and use an ensemble of 4 dynamics models to implement USAD as described in Section 4.2. We parameterize the policy using a 2-layer tanh-MLP, and train it using model-based NPG [28]. We evaluate the learned policies using rollouts in the (real) environment, but these rollouts are not made available to the algorithm in any way for purposes of learning. This is similar to evaluation protocols followed in prior work [18, 15, 16]. We present all our results averaged over 5 different random seeds. Note that we use the same hyperparameters for all random seeds. In contrast, the prior works whose results we compare against tune hyper-parameters separately for each random seed [18].

**Comparison of `MOReL`'s performance with prior work**   We compare `MOReL` with prior SOTA algorithms like BCQ, BEAR, and all variants of BRAC. The results are summarized in Table 1. For fairness of comparison, we reproduce results from prior work and do not run the algorithms ourselves, since random-seed-specific hyperparameter tuning is required to achieve the results reported by prior work [18]. We provide a more expansive table with additional baseline algorithms in the appendix. Our algorithm, `MOReL`, achives SOTA results in 12 out of the 20 environment-dataset combinations, overlaps in error bars for 3 other combinations, and is competitive in the remaining cases. In contrast, the next best approach (a BRAC variant) achieves SOTA results in 5 out of 20 configurations.

Table 1: Results in various environment-exploration combinations. Baselines are reproduced from Wu et al. [18]. Prior work does not provide error bars. For MOReL results, error bars indicate the standard deviation across 5 different random seeds. We choose SOTA result based on the average performance.

| Environment: Ant-v2 | | | | | |
|---|---|---|---|---|---|
| Algorithm | BCQ [15] | BEAR [16] | BRAC [18] | Best Baseline | MOReL (Ours) |
| Pure | 1921 | 2100 | 2839 | 2839 | **3663±247** |
| Eps-1 | 1864 | 1897 | 2672 | 2672 | **3305±413** |
| Eps-3 | 1504 | 2008 | 2602 | 2602 | **3008±231** |
| Gauss-1 | 1731 | 2054 | 2667 | 2667 | **3329±270** |
| Gauss-3 | 1887 | 2018 | 2640 | 2661 | **3693±33** |

| Environment: Hopper-v2 | | | | | |
|---|---|---|---|---|---|
| Algorithm | BCQ [15] | BEAR [16] | BRAC [18] | Best Baseline | MOReL (Ours) |
| Pure | 1543 | 0 | 2291 | 2774 | **3642±54** |
| Eps-1 | 1652 | 1620 | 2282 | 2360 | **3724±46** |
| Eps-3 | 1632 | 2213 | 1892 | 2892 | **3535±91** |
| Gauss-1 | 1599 | 1825 | 2255 | 2255 | **3653±52** |
| Gauss-3 | 1590 | 1720 | 1458 | 2097 | **3648±148** |

| Environment: HalfCheetah-v2 | | | | | |
|---|---|---|---|---|---|
| Algorithm | BCQ [15] | BEAR [16] | BRAC [18] | Best Baseline | MOReL (Ours) |
| Pure | 5064 | 5325 | 6207 | **6209** | 6028±192 |
| Eps-1 | 5693 | 5435 | **6307** | **6307** | 5861±192 |
| Eps-3 | 5588 | 5149 | 6263 | 6359 | 5869±139 |
| Gauss-1 | 5614 | 5394 | **6323** | 6323 | 6026±74 |
| Gauss-3 | 5837 | 5329 | **6400** | 6400 | 5892±128 |

| Environment: Walker-v2 | | | | | |
|---|---|---|---|---|---|
| Algorithm | BCQ [15] | BEAR [16] | BRAC [18] | Best Baseline | MOReL (Ours) |
| Pure | 2095 | 2646 | 2694 | 2907 | **3709±159** |
| Eps-1 | 1921 | 2695 | 3241 | **3490** | 2899±588 |
| Eps-3 | 1953 | 2608 | 3255 | 3255 | **3186±92** |
| Gauss-1 | 2094 | 2539 | 2893 | 3193 | **4027±314** |
| Gauss-3 | 1734 | 2194 | **3368** | 3368 | 2828±589 |

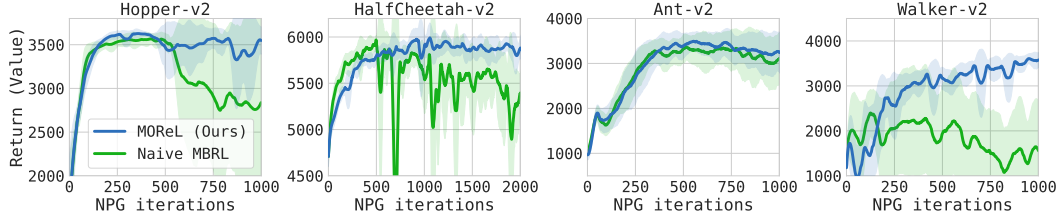

Figure 3: MOReL and Naive MBRL learning curves. The x-axis plots the number of model-based NPG iterations, while y axis plots the return (value) in the real environment. The naive MBRL algorithm is highly unstable while MOReL leads to stable and near-monotonic learning. Notice however that even naive MBRL learns a policy that performs often as well as the best model-free offline RL algorithms.

**Quality of logging policy** Section 4.1 indicates that it is not possible for any offline RL algorithm to learn a near-optimal policy when faced with support mismatch between the dataset and optimal policy. To verify this experimentally for MOReL, we consider two datasets (of the same size) collected using the Pure strategy. The first uses a partially trained policy $\pi_p$ (called Pure-partial), which is the same as the Pure dataset studied in Table 1. The second dataset is collected using an untrained random Gaussian policy (called Pure-random). Table 2 compares the results of MOReL using these two datasets. We observe that the value of policy learned with Pure-partial dataset far exceeds the value with the Pure-random dataset. Thus, the value of policy used for data logging plays a crucial role in the performance achievable with offline RL.

Table 2: Value of the policy learned by MOReL (5 random seeds) when working with a dataset collected with a random (untrained) policy (Pure-random) and a partially trained policy (Pure-partial).

| Environment | Pure-random | Pure-partial |
|---|---|---|
| Hopper-v2 | $2354 \pm 443$ | $3642 \pm 54$ |
| HalfCheetah-v2 | $2698 \pm 230$ | $6028 \pm 192$ |
| Walker2d-v2 | $1290 \pm 325$ | $3709 \pm 159$ |
| Ant-v2 | $1001 \pm 3$ | $3663 \pm 247$ |

### Importance of Pessimistic MDP

To highlight the importance of P-MDP, we consider the Pure-partial dataset outlined above. We compare MOReL with a naïve MBRL approach that first learns a dynamics model using the offline data, followed by running model-based NPG without any safeguards against model inaccuracy. The results are summarized in Figure 3. We observe the naïve MBRL approach already works well, achieving comparable results to prior methods like BCQ and BEAR. However, MOReL clearly exhibits more stable and monotonic learning progress. This is particularly evident in Hopper-v2, HalfCheetah-v2, and Walker2d-v2, where an uncoordinated set of actions can result in the agent falling over. Furthermore, in the case of naïve MBRL, we observe that performance can quickly degrade after a few hundred steps of policy improvement, such as in case of Hopper-v2, HalfCheetah-v2. This suggests that the learned model is being over-exploited. In contrast, with MOReL, we observe that the learning curve is stable and nearly monotonic even after many steps of policy optimization.

### Transfer from P-MDP to environment

Finally, we study how the learning progress in P-MDP relates to the progress in the environment. Our theoretical results (Theorem 1) suggest that the value of a policy in the P-MDP cannot substantially exceed the value in the environment. This makes the value in the P-MDP an approximate lower bound on the true performance, and a good surrogate for optimization. In Figure 4, we plot the value (return) of the policy in the P-MDP and environment over the course of learning. Note that the policy is being learned in the P-MDP, and as a result we observe a clear monotonic learning curve for value in the P-MDP, consistent with the monotonic improvement theory of policy gradient methods [75, 76]. We observe that the value in the true environment closely correlates with the value in P-MDP. In particular, the P-MDP value never substantially exceeds the true performance, suggesting that the pessimism helps to avoid model exploitation.

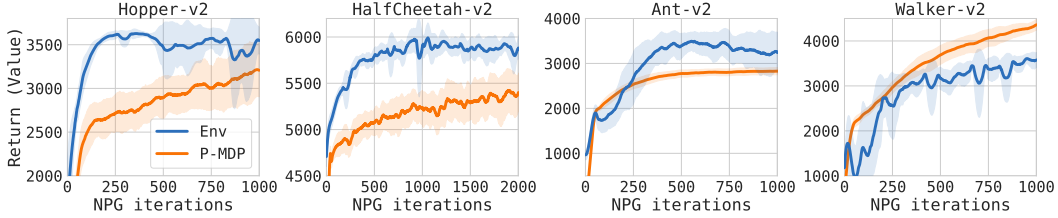

Figure 4: Learning curve using the `Pure-partial` dataset, see paper text for details. The policy is learned using the pessimistic MDP (P-MDP), and we plot the performance in both the P-MDP and the real environment over the course of learning. We observe that the performance in the P-MDP closely tracks the true performance and never substantially exceeds it, as predicted in section 4.1. This shows that the policy value in the P-MDP serves as a good surrogate for the purposes of offline policy evaluation and learning.

## 6  Conclusions

We introduced `MOReL`, a new model-based framework for offline RL. `MOReL` incorporates both *generalization* and *pessimism* (or conservatism). This enables `MOReL` to perform policy improvement in known states that may not directly occur in the static offline dataset, but can nevertheless be predicted using the dataset by leveraging the power of generalization. At the same time, due to the use of pessimism, `MOReL` ensures that the agent does not drift to unknown states where the agent cannot predict accurately using the static dataset.

Theoretically, we obtain bounds on the suboptimality of `MOReL` which improve over those in prior work. We further showed that this suboptimality bound cannot be improved upon by *any* offline RL algorithm in the worst case. Experimentally, we evaluated `MOReL` in the standard continuous control benchmarks in OpenAI gym and showed that it achieves state of the art results. The modular structure of `MOReL` comprising of model learning, uncertainty estimation, and model-based planning allows the use of a variety of approaches such as multi-step prediction for model learning, abstention for uncertainty estimation, or model-predictive control for action selection. In future work, we hope to explore these directions.

## Acknowledgements And Funding Disclosure

The authors thank Prof. Emo Todorov for generously providing the MuJoCo simulator for use in this paper. Rahul Kidambi thanks Mohammad Ghavamzadeh and Rasool Fakoor for pointers to related works and other valuable discussions/pointers about offline RL. Aravind Rajeswaran thanks Profs. Sham Kakade and Emo Todorov for valuable discussions. The authors also thank Prof. Nan Jiang and Anirudh Vemula for pointers to related work. Rahul Kidambi acknowledges funding from NSF Award CCF − 1740822 and computing resources from the Cornell "Graphite" cluster. Part of this work was completed when Aravind held dual affiliations with the University of Washington and Google Brain. Aravind acknowledges financial support through the JP Morgan PhD Fellowship in AI. Thorsten Joachims acknowledges funding from NSF Award IIS − 1901168. All content represents the opinion of the authors, which is not necessarily shared or endorsed by their respective employers and/or sponsors.

## Broader Impact

This paper studies offline RL, which allows for data driven policy learning using pre-collected datasets. The ability to train policies offline can expand the range of applications where RL can be applied as well as the sample efficiency of any downstream online learning. Since the dataset has already been collected, offline RL enables us to abstract away the exploration or data collection challenge. Safe exploration is crucial for applications like robotics and healthcare, where poorly designed exploratory actions can have harmful physical consequences. Avoiding online exploration by an autonomous agent, and working with a safely collected dataset, can have the broader impact of alleviating safety challenges in RL. That said, the impact of RL agents to the society at large is highly dependent on the

design of the reward function. If the reward function is designed by malicious actors, any RL agent, be it offline or not, can present negative consequences. Therefore, the design of reward functions requires checks, vetting, and scrutiny to ensure RL algorithms are aligned with societal norms.

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
