[Supplementary Material]

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

# A   Theoretical Results: Proofs For Section 4.1

In this section, we present the proofs of our main results Theorem 1 and Proposition 4.

*Proof of Theorem 1.* We wish to show the following two inequalities.

$$J_{\hat{\rho}_0}(\pi, \hat{\mathcal{M}}_p) \geq J_{\rho_0}(\pi, \mathcal{M}) - \frac{2R_{\max}}{1-\gamma} \cdot D_{TV}(\rho_0, \hat{\rho}_0) - \frac{2\gamma R_{\max}}{(1-\gamma)^2} \cdot \alpha - \frac{2R_{\max}}{1-\gamma} \cdot \mathbb{E}\left[\gamma^{T_u^\pi}\right], \text{ and}$$

$$J_{\hat{\rho}_0}(\pi, \hat{\mathcal{M}}_p) \leq J_{\rho_0}(\pi, \mathcal{M}) + \frac{2R_{\max}}{1-\gamma} \cdot D_{TV}(\rho_0, \hat{\rho}_0) + \frac{2\gamma R_{\max}}{(1-\gamma)^2} \cdot \alpha.$$

The proof of this theorem is inspired by the simulation lemma of [55], with some additional modifications due to pessimism, and goes through the pessimistic MDP $\mathcal{M}_p$, which is the same as $\hat{\mathcal{M}}_p$ except that the starting state distribution is $\rho_0$ instead of $\hat{\rho}_0$ and the transition probability from a known state-action pair $(s, a)$ is $P(s'|s, a)$ instead of $\widehat{P}(s'|s, a)$. More concretely, $\mathcal{M}_p$ is described by $\{S \cup \text{HALT}, A, r_p, P_p, \rho_0, \gamma\}$, where HALT is an additional absorbing state we introduce similar to what we did for $\hat{\mathcal{M}}_p$. The modified reward and transition dynamics are given by:

$$P_p(s'|s, a) = \begin{cases} \delta(s' = \text{HALT}) & \text{if } U^\alpha(s, a) = \text{TRUE} \\ & \text{or } s = \text{HALT} \\ P(s'|s, a) & \text{otherwise.} \end{cases} \qquad r_p(s, a) = \begin{cases} -\kappa & \text{if } s = \text{HALT} \\ r(s, a) & \text{otherwise} \end{cases}$$

We first show that

$$J_{\hat{\rho}_0}(\pi, \hat{\mathcal{M}}_p) \geq J_{\rho_0}(\pi, \mathcal{M}_p) - \frac{2R_{\max}}{1-\gamma} \cdot D_{TV}(\rho_0, \hat{\rho}_0) - \frac{2\gamma R_{\max}}{(1-\gamma)^2} \cdot \alpha, \text{ and}$$

$$J_{\hat{\rho}_0}(\pi, \hat{\mathcal{M}}_p) \leq J_{\rho_0}(\pi, \mathcal{M}_p) + \frac{2R_{\max}}{1-\gamma} \cdot D_{TV}(\rho_0, \hat{\rho}_0) + \frac{2\gamma R_{\max}}{(1-\gamma)^2} \cdot \alpha,$$

The main idea is to couple the evolutions of any given policy on the pessimistic MDP $\mathcal{M}_p$ and the model-based pessimistic MDP $\hat{\mathcal{M}}_p$ so that $(s_{t-1}, a_{t-1}) \stackrel{\text{def}}{=} (s_{t-1}^{\mathcal{M}_p}, a_{t-1}^{\mathcal{M}_p}) = (s_{t-1}^{\hat{\mathcal{M}}_p}, a_{t-1}^{\hat{\mathcal{M}}_p})$.

Assuming that such a coupling can be performed in the first step, since $\left\| P(s, a) - \hat{P}(s, a) \right\|_1 \leq \alpha$, this coupling can be performed at each subsequent step with probability $1 - \alpha$. The probability that the coupling is not valid at time $t$ is at most $1 - (1 - \alpha)^t$. So the total difference in the values of the policy $\pi$ on the two MDPs can be upper bounded as:

$$\left| J_{\hat{\rho}_0}(\pi, \hat{\mathcal{M}}_p) - J_{\rho_0}(\pi, \mathcal{M}_p) \right| \leq \frac{2R_{\max}}{1-\gamma} \cdot D_{TV}(\rho_0, \hat{\rho}_0) + \sum_t \gamma^t \left(1 - (1-\alpha)^t\right) \cdot 2 \cdot R_{\max}$$

$$\leq \frac{2R_{\max}}{1-\gamma} \cdot D_{TV}(\rho_0, \hat{\rho}_0) + \frac{2\gamma R_{\max}}{(1-\gamma)^2} \cdot \alpha.$$

We now argue that

$$J_{\rho_0}(\pi, \mathcal{M}_p) \geq J_{\rho_0}(\pi, \mathcal{M}) - \frac{2R_{\max}}{1-\gamma} \cdot \mathbb{E}\left[\gamma^{T_u^\pi}\right], \text{ and}$$

$$J_{\rho_0}(\pi, \mathcal{M}_p) \leq J_{\rho_0}(\pi, \mathcal{M}).$$

For the first part, we see that the evolution of any policy $\pi$ on the pessimistic MDP $\mathcal{M}_p$, can be coupled with the evolution of $\pi$ on the actual MDP $\mathcal{M}$ until $\pi$ encounters an unknown state. From this point, the total rewards obtained on the pessimistic MDP $\mathcal{M}_p$ will be $\frac{-R_{\max}}{1-\gamma}$, while the maximum total reward obtained by $\pi$ on $\mathcal{M}$ from that point on is $\frac{R_{\max}}{1-\gamma}$. Multiplying by the discount factor $\mathbb{E}\left[\gamma^{T_u^\pi}\right]$ proves the first part.

For the second part, consider any policy $\pi$ and let it evolve on the MDP $\mathcal{M}$ as $(s, a, s'_{\mathcal{M}})$. Simulate an evolution of the same policy $\pi$ on $\mathcal{M}_p$, $\left(s, a, s'_{\mathcal{M}_p}\right)$, as follows: if $(s, a) \in SA_k$, then $s'_{\mathcal{M}_p} = s'_{\mathcal{M}}$ and if $(s, a) \in \mathcal{U}$, then $s'_{\mathcal{M}_p} = \text{HALT}$. We see that the rewards obtained by $\pi$ on each transition in $\mathcal{M}_p$ is less than or equal to that obtained by $\pi$ on the same transition in $\mathcal{M}$. This proves the second part of the lemma. $\square$

Figure 5: This example shows that the suboptimality of any offline RL algorithm is at least $\frac{R_{\max}}{4(1-\gamma)^2} \cdot \frac{d^{\pi^*,\mathcal{M}}(\mathcal{U}_D)}{\log\frac{1}{1-\gamma}}$ in the worst case and hence Corollary 3 is tight. The states $1, 2, \cdots, k+1$ in the MDP are depicted under the circles. The actions $a_1, a_2, a_3$, rewards and transitions are depicted on the arrows connecting the states. The actions taken by the behavior (i.e. the data collection) policy are depicted in blue. See Proposition 4 and its proof for more details.

*Proof of Corollary 2.* By the suboptimality assumption on the planning algorithm and Theorem 1, we have

$$
\begin{aligned}
J_{\rho_0}(\pi_{\text{out}}, \mathcal{M}) &\geq J_{\hat{\rho}_0}(\pi_{\text{out}}, \hat{\mathcal{M}}_p) - \frac{2R_{\max}}{1-\gamma} \cdot D_{TV}(\rho_0, \hat{\rho}_0) - \frac{2\gamma R_{\max}}{(1-\gamma)^2} \cdot \alpha \\
&\geq J_{\hat{\rho}_0}(\pi^*, \hat{\mathcal{M}}_p) - \epsilon_\pi - \frac{2R_{\max}}{1-\gamma} \cdot D_{TV}(\rho_0, \hat{\rho}_0) - \frac{2\gamma R_{\max}}{(1-\gamma)^2} \cdot \alpha \\
&\geq J_{\rho_0}(\pi^*, \mathcal{M}) - \epsilon_\pi - \frac{4R_{\max}}{1-\gamma} \cdot D_{TV}(\rho_0, \hat{\rho}_0) - \frac{4\gamma R_{\max}}{(1-\gamma)^2} \cdot \alpha - \frac{2R_{\max}}{1-\gamma} \cdot \mathbb{E}\left[\gamma^{T_{\mathcal{U}}^{\pi^*}}\right].
\end{aligned}
$$

This proves the result. □

**Lemma 5.** *(Hitting time and visitation distribution) For any set $\mathcal{S} \subseteq S \times A$, and any policy $\pi$, we have $\mathbb{E}\left[\gamma^{T_{\mathcal{S}}^\pi}\right] \leq \frac{1}{1-\gamma} \cdot d^{\pi,\mathcal{M}}(\mathcal{S})$.*

*Proof of Lemma 5.* The proof is rather straightforward. We have

$$
\begin{aligned}
\mathbb{E}\left[\gamma^{T_{\mathcal{U}}^\pi}\right] &\leq \sum_{(s',a')\in\mathcal{U}} \mathbb{E}\left[\gamma^{T_{(s',a')}^\pi}\right] \leq \sum_{(s',a')\in\mathcal{U}} \sum_{t=0}^\infty \gamma^t P(s_t = s', a_t = a'|s_0 \sim \rho_0, \pi, \mathcal{M}) \\
&= \frac{1}{1-\gamma} \sum_{(s',a')\in\mathcal{U}} d^{\pi,\mathcal{M}}(s', a') = \frac{1}{1-\gamma} \cdot d^{\pi,\mathcal{M}}(\mathcal{U}).
\end{aligned}
$$

□

*Proof of Proposition 4.* We consider the MDP in Figure 5, where we set $k = 10\log\frac{1}{1-\gamma}$. The MDP has $k+1$ states, with three actions $a_1, a_2$ and $a_3$ at each state. The rewards (shown on the transition arrows) are all $0$ except for the action $a_1$ taken in state $k+1$, in which case it is $1$. Note that the rewards can be scaled by $R_{\max}$ but for simplicity, we consider the setting with $R_{\max} = 1$. It is clear that the optimal policy $\pi^*$ is to take the action $a_1$ in all the states. The starting state distribution $\rho_0$ is state $1$ with probability $p_0 \stackrel{\text{def}}{=} \frac{\epsilon}{(1-\gamma)\log\frac{1}{1-\gamma}}$ and state $k+1$ with probability $1 - p_0$. The actions taken by the data collection policy are shown in blue. Since the dataset consists only of (state, action, reward, next state) pairs $(1, a_1, 0, 2), (2, a_2, 0, 1)$ and $(k+1, a_1, 1, k+1)$ we see that $\mathcal{U}_D = (S \times A) \setminus \{(1, a_1), (2, a_2), (k+1, a_1)\}$ and $d^{\pi^*,\mathcal{M}}(\mathcal{U}_D) = (1-\gamma) \cdot \sum_{t=1}^{k-1} \gamma^t \cdot p_0 \leq (1-\gamma) \cdot (k-1) \cdot p_0 \leq \epsilon$ proving the first claim. Since none of the states and actions in $\mathcal{U}_D$ are seen in the dataset, after permuting the actions if necessary, the expected time taken by any policy learned from the dataset, to reach state $k+1$ starting from state $1$ is at least $\exp(k/5) \geq (1-\gamma)^{-2}$. So, the value of any policy $\hat{\pi}$ learned from the dataset is at most $\frac{1-p_0}{1-\gamma} + \frac{p_0 \cdot \gamma^{(1-\gamma)^{-2}}}{1-\gamma} = \frac{1}{1-\gamma} - p_0 \cdot \frac{1-\gamma^{(1-\gamma)^{-2}}}{1-\gamma} \leq \frac{1}{1-\gamma} - \frac{3p_0}{4(1-\gamma)}$, where we used $\gamma \in [0.95, 1)$ in the last step. On the other hand, the value of $\pi^*$ is at least $\frac{1-p_0}{1-\gamma} + p_0 \cdot \left(\frac{1}{1-\gamma} - k\right)$. So the suboptimality

of any learned policy is at least $p_0 \cdot \left( \frac{3}{4(1-\gamma)} - k \right) = p_0 \cdot \left( \frac{3}{4(1-\gamma)} - 10 \log \frac{1}{1-\gamma} \right) \geq \frac{p_0}{4(1-\gamma)}$, where we again used $\gamma \in [0.95, 1)$ in the last step. Substituting the value of $p_0$ proves the proposition. $\square$

# B  Detailed Related Work

## B.1  Offline RL

Offline RL dates at least to the work of Lange et al. [12]. In this setting, an RL agent is provided access to a typically large offline dataset, using which it has to produce a highly rewarding policy. This has direct applications in fields like healthcare [33, 34, 35], recommendation systems [36, 37, 38, 39], dialogue systems [40, 19, 41], and autonomous driving [42]. We refer the readers to the review paper of Levine et al. [77] for an overview of potential applications. On the algorithmic front, prior work in offline RL can be broadly categorized into three groups as described below.

**Importance sampling**    The first approach to offline RL is through importance sampling. In this approach, trajectories from the offline dataset are directly used to estimate the policy gradient, which is subsequently corrected using importance weights. This approach is particularly common in contextual bandits literature [43, 36, 37] where the importance weights are relatively easier to estimate due to the non-sequential nature of the problem. For MDPs, Liu et al. [44] present an importance sampling based off-policy policy gradient method by estimating state distribution weights [45, 46, 47]. The work of Liu et al. [44] also utilizes the notion of *pessimism* by optimizing only over a subset of states visited by the behavioral policy. They utilize importance weighted policy gradient (with estimated importance weights) to optimize this MDP. However, their work does not naturally capture a notion of generalization over the state space. Moreover, their results require strong assumptions on the data collecting policy in the sense of ensuring support on states visited by the optimal policy. Our framework, MOReL, provides the same guarantees under identical assumptions, but we also show that the performance of MOReL degrades gracefully when these assumptions aren't satisfied.

**Dynamic programming**    The overwhelming majority of recent algorithmic work in offline RL is through the paradigm of approximate dynamic programming. In principle, any off-policy algorithm based on Q-learning [48, 49] or actor-critic architectures [50, 78, 51] can be used with a static offline dataset. However, recent empirical studies confirm that such a direct extension leads to poor results due to the challenges of overestimation bias in generalization and distribution shift. To address overestimation bias, prior work has proposed approaches like ensembles of Q-networks [17, 15, 19]. As for distribution shift, the principle approach used is to regularize the learned policy towards the data logging policy [15, 16, 18]. Different regularization schemes, such as those based on KL-divergence and maximum mean discrepancy (MMD), have been considered in the past. Wu et al. [18] perform a comparative study of such regularization schemes and find that they all perform comparably. ADP-based offline RL has also be studied theoretically [26, 52], with Chen and Jiang [52] providing an information-theoretic lower bound on sample complexity. However, these works again don't study the impact of support mismatch between logging policy and optimal policy. Finally, a recent line of work [21, 79] focuses on obtaining provably convergent methods for minimizing the (one-step) Bellman error using Duality theory. While they show promising results in continuous control tasks in the online RL setting, their performance in the offline RL setting is yet to be studied.

**Model-based RL**    The interplay between model-based methods and offline RL has only been sparsely explored. The work of Ross & Bagnell [32] theoretically studied the performance of MBRL in the batch setting. In particular, the algorithm they analyzed involves learning a dynamics model using the offline dataset, and subsequently planning in the learned model without any additional safeguards. Their theoretical results are largely negative for this algorithm, suggesting that in the worst case, this algorithm could have arbitrarily large sub-optimality. In addition, their sub-optimality bounds become pathologically loose when the data logging distribution does not share support with the distribution of the optimal policy. Model-based offline RL methods from a safe policy improvement perspective have also been considered [53]. In contrast to both these works, we present a novel algorithmic framework that constructs and pessimistic MDP, and show that this is crucial for better empirical results and sharper theoretical analysis.

## B.2 Advances in Model-Based RL

Since our work utilizes model-based RL, we review the most directly related work in the online RL setting. Classical works in MBRL have focused extensively on tabular MDPs and linear quadratic regulartor (LQR). For tabular MDPs (in the online RL setting), the first known polynomial time algorithms were the model-based algorithms of $E^3$ [55] and R-MAX [56]. More recent work suggests that model-based methods are minimax optimal for tabular MDPs when equipped with a wide restart state distribution [80]. However, these works critically rely on the tabular nature of the problem. Since each table entry is typically considered to be independent, and updates to any entry to do not affect other entries, tabular MDPs do not afford any notion of generalization. The metric-$E^3$ [57] algorithm aims to overcome this challenge by considering an underlying metric space for state-actions that enables generalization. While this work provides a strong theoretical basis, it does not directly provide a practical algorithm that can be used with function approximation. Our work is perhaps conceptually closest to $E^3$ and metric-$E^3$ which partitions the state space into known and unknown regions. A cornerstone of MOReL is the P-MDP which partitions the state space into known and unknown regions, as in, $E^3$ [55] and R-MAX [56], but these constructions were not developed to encourage pessimism. However, all of these works primarily deal with the standard (online) RL setting. Our work differs in its focus on offline RL, where we show the P-MDP construction plays a crucial role. Moreover, direct practical instantiations of $E^3$ and metric-$E^3$ with function approximation have remained elusive.

In recent years, along with an explosion of interest in deep RL, MBRL has emerged as a powerful class of approaches for sample efficient learning. Modern MBRL methods (typically in the online RL setting) can support the use of flexible function approximators like neural networks, as well as generic priors like smoothness and approximate knowledge of physics [22], enabling the learning of accurate models. Furthermore, MBRL can draw upon the rich literature on model-based planning including model predictive control (MPC) [23, 24, 64, 72], search based planning [25, 65], dynamic programming [26, 81], and policy optimization [82, 76, 66, 27, 51]. These advances in MBRL have enabled highly sample efficient learning in widely studied benchmark tasks [83, 29, 84, 85, 28], as well as in a number of challenging robotic control tasks like aggressive driving [64], dexterous hand manipulation [86, 28], and quadrupedal locomotion [87]. Among these works, the recent work of Rajeswaran et al. [28] demonstrated state of the art results with MBRL in a range of benchmark tasks, and forms the basis for our practical implementation. In particular, our model learning and policy optimization subroutines are extended from the MAL framework in Rajeswaran et al. [28]. However, our work crucially differs from it due to the pessimistic MDP construction, which we show is important for success in the offline RL setting.

# C  Additional Experimental Details And Setup

## C.1  Environment Details And Setup

As mentioned before, following recent efforts in offline RL [15, 16, 18], we consider four continuous control tasks: Hopper-v2, HalfCheetah-v2, Ant-v2, Walker2d-v2 from OpenAI gym [73] simulated with MuJoCo [74]. As normally done in MBRL literature with OpenAI gym tasks [30, 88, 89, 28], we reduce the planning horizon for the environments to 400 or 500. Similar to [89, 28], we append our state parameterization with center of mass velocity to compute the reward from observations. Mirroring realistic settings, we assume access to data collected using a partially trained (sub-optimal) policy interacting with the environment. To obtain a partially trained policy $\pi_p$ [15, 16, 18], we run (online) TRPO [76] until the policy reaches a value of 1000, 4000, 1000, 1000 respectively for these environments. This policy in conjunction with exploration strategies are used to collect the datasets (see below for more details). All our results are obtained by averaging runs of five random seeds (for the planning algorithm), with the seed values being $123, 246, 369, 492, 615$. Each of our experiments are run with 1 NVidia GPU and 2 CPUs using a total of 16GB of memory.

## C.2  Dynamics Model, Policy Network And Evaluation

We use 2 hidden layer MLPs with 512 (for Hopper-v2, Walker2d-v2, Ant-v2) or 1024 (for HalfCheetah-v2) ReLU activated nodes each for representing the dynamics model, use an ensemble of four such models for building the USAD, and our policy is represented with a 2 hidden layer MLP with 32 tanh activated nodes in each layer. The dynamics model is learnt using Adam [68]

and the policy parameters are learnt using model-based NPG steps [28]. We set hyper-parameters and track policy learning curve by performing rollouts in the real environment; these rollouts aren't used for other purposes in the learning procedure. Similar protocols are used in prior work[15, 16, 18].

### C.3 Description Of Types Of Policies

We build off the experimental setup of [18]. Towards this, we first go over some notation. Firstly, let $\pi_b$ represent the behavior policy, $\pi_r$ is a random policy that picks actions according to a certain probability distribution (for e.g., Gaussian $\pi_r^g$/Uniform $\pi_r^u$ etc.), $\pi_p$ a partially-trained policy, which one can assume is better than a random policy in value. Let $\pi_b^u(q)$ be a policy that plays random actions with probability $q$, and sampled actions from $\pi_b$ with probability $1 - q$. Let $\pi_b^g(\beta)$ be a policy that adds zero-mean Gaussian noise with standard deviation $\beta$ to actions sampled from $\pi_b$. Consider a behavior policy which, for instance, can be a partially trained data logging policy $\pi_b$. We consider five different exploration strategies, each corresponding to adding different kinds of exploratory noise to $\pi_b$, as described below.

### C.4 Datasets And Exploration Strategies

For each environment, we use a combination of a behavior policy $\pi_b$, a noisy behavior policy $\tilde{\pi}_b$ (see below), and a pure random stochastic process $\pi_r$ to collect several datasets, following Wu et al. [18]. Each dataset contains the equivalent of 1 million timesteps of interactions with the environment. See below for detailed instructions.

($\mathcal{E}1$) `Pure`: The entire dataset is collected with the data logging (behavioral) policy $\pi_b$.

($\mathcal{E}2$) `Eps-1`: 40% of the dataset is collected with $\pi_b$, another 40% collected with $\pi_b^u(0.1)$, and the final 20% is collected with a random policy $\pi_r$.

($\mathcal{E}3$) `Eps-3`: 40% of the dataset is collected with $\pi_b$, another 40% collected with $\pi_b^u(0.3)$, and the final 20% is collected with a random policy $\pi_r$.

($\mathcal{E}4$) `Gauss-1`: 40% of the dataset is collected with $\pi_b$, another 40% collected with $\pi_b^g(0.1)$, and the final 20% is collected with a random policy $\pi_r$.

($\mathcal{E}5$) `Gauss-3`: 40% of the dataset is collected with $\pi_b$, another 40% collected with $\pi_b^g(0.3)$, and the final 20% is collected with a random policy $\pi_r$.

### C.5 Hyperparameter Selection

Refer to table 3 for details with regards to parameters of `MOReL`. For all environments and data collection strategies, we learn two-layer MLP based dynamics models with ReLU activations by minimizing the one-step prediction errors using Adam [68] and utilize four of these models for defining the USAD. The negative reward for defining the absorbing unknown state is set as the minimum reward in the dataset $\mathcal{D}$ offsetted by a value that is searched over $\{30, 50, 100, 200\}$.

**Ascertaining unknown state-action pairs:** In order to ascertain unknown state-action pairs, we compute the model disagreement as: $\mathrm{disc}(s, a) = \max_{i \neq j} ||f_{\phi_i}(s, a) - f_{\phi_j}(s, a)||_2$, where, $f_{\phi_i}$ and $f_{\phi_j}$ are members of the ensemble of learnt dynamics model. Specifically, we compute $\mathrm{disc}(s, a)$ over all state-action pairs that occur in the static dataset $\mathcal{D}$. Next, we can compute the mean $\mu_d$, standard deviation $\sigma_d$ and the max $m_d$ of the disagreements evaluated for every state-action pair occuring in the dataset. Then, we utilize an upper-confidence inspired strategy by defining a threshold $\mathrm{thresh} = \mu_d + \beta \cdot \sigma_d$. The value of beta is tuned between 0 to $\beta_{\max} = (m_d - \mu_d)/\sigma_d$ in steps of 5. For any model-based rollout encountered during planning, if the discrepancy of the state-action pair at a given timestep exceeds thresh, the rollout is truncated at this timestep and is assigned a large negative reward. We emphasize that for every environment, all hyper-parameters (except for $\beta$) is maintained at the same value across all exploration settings.

With regards to the policy and the planning algorithm, we consider a $(32, 32)$ tanh MLP optimized using normalized model-based NPG steps (see, for instance, the work of Rajeswaran et al. [28] for the model-based NPG algorithm). Parameters of model-based NPG is described in table 4.

Table 3: Hyper-parameters for each environment for `MOReL`. Note that most hyper-parameters are common across domains, and the differences are primarily in reward penalty and number of fitting epochs, which are necessarily environment specific.

| Environment: Ant-v2 | |
|---|---|
| **Parameter** | **Value** |
| Dynamics Model | MLP(512, 512) |
| Activation | ReLU |
| # Training Epochs | 300 |
| Adam Stepsize | 5e-4 |
| Batch Size | 256 |
| Horizon | 500 |
| Negative Reward | $r_{\min}(\mathcal{D}) - 100$ |
| USAD | 4-dynamics models |

| Environment: Hopper-v2 | |
|---|---|
| **Parameter** | **Value** |
| Dynamics Model | MLP(512, 512) |
| Activation | ReLU |
| # Training Epochs | 300 |
| Adam Stepsize | 5e-4 |
| Batch Size | 256 |
| Horizon | 400 |
| Negative Reward | $r_{\min}(\mathcal{D}) - 50$ |
| USAD | 4-dynamics models |

| Environment: HalfCheetah-v2 | |
|---|---|
| **Parameter** | **Value** |
| Dynamics Model | MLP(1024,1024) |
| Activation | ReLU |
| # Training Epochs | 3000 |
| Adam Stepsize | 5e-4 |
| Batch Size | 256 |
| Horizon | 500 |
| Negative Reward | $r_{\min}(\mathcal{D}) - 200$ |
| USAD | 4-dynamics models |

| Environment: Walker-v2 | |
|---|---|
| **Parameter** | **Value** |
| Dynamics Model | MLP(512, 512) |
| Activation | ReLU |
| # Training Epochs | 300 |
| Adam Stepsize | 5e-4 |
| Batch Size | 256 |
| Horizon | 400 |
| Negative Reward | $r_{\min}(\mathcal{D}) - 30$ |
| USAD | 4-dynamics models |

Table 4: Hyper-parameters for model-based policy optimization. Note that most hyperparameters are common except the number of iterations and exploration noise.

| Environment: Ant-v2 | |
|---|---|
| **Parameter** | **Value** |
| Policy Net | MLP(32,32) |
| Non-linearity | Tanh |
| # updates | 1000 |
| $\log \sigma_{\text{init}}$ | -0.1 |
| $\log \sigma_{\text{min}}$ | -2.0 |
| # trajectories for gradient | 200 |
| # Eval trajectories | 25 |
| # CG Steps/Damping | 10, 1e-4 |

| Environment: Hopper-v2 | |
|---|---|
| **Parameter** | **Value** |
| Policy Net | MLP(32,32) |
| Non-linearity | Tanh |
| # updates | 500 |
| $\log \sigma_{\text{init}}$ | -0.25 |
| $\log \sigma_{\text{min}}$ | -2.0 |
| # trajectories for gradient | 50 |
| # Eval trajectories | 25 |
| # CG Steps/Damping | 25, 1e-4 |

| Environment: HalfCheetah-v2 | |
|---|---|
| **Parameter** | **Value** |
| Policy Net | MLP(32,32) |
| Non-linearity | Tanh |
| # updates | 2500 |
| $\log \sigma_{\text{init}}$ | -1.0 |
| $\log \sigma_{\text{min}}$ | -2.0 |
| # trajectories for gradient | 40 |
| # Eval trajectories | 25 |
| # CG Steps/Damping | 10, 1e-4 |

| Environment: Walker-v2 | |
|---|---|
| **Parameter** | **Value** |
| Policy Net | MLP(32,32) |
| Non-linearity | Tanh |
| # updates | 1000 |
| $\log \sigma_{\text{init}}$ | -0.5 |
| $\log \sigma_{\text{min}}$ | -2.0 |
| # trajectories for gradient | 100 |
| # Eval trajectories | 25 |
| # CG Steps/Damping | 25, 1e-4 |

Table 5: Results in the four environments and five exploration configurations.

| Environment: Ant-v2 | | | Partially trained policy: 1241 | | |
|---|---|---|---|---|---|
| Algorithm | Pure ($\mathcal{E}1$) | Eps-1 ($\mathcal{E}2$) | Eps-3 ($\mathcal{E}3$) | Gauss-1 ($\mathcal{E}4$) | Gauss-3 ($\mathcal{E}5$) |
| SAC [51] | 0 | -1109 | -911 | -1071 | -1498 |
| BC | 1235 | 1300 | 1278 | 1203 | 1240 |
| BCQ [15] | 1921 | 1864 | 1504 | 1731 | 1887 |
| BEAR [16] | 2100 | 1897 | 2008 | 2054 | 2018 |
| MMD_vp [18] | 2839 | 2672 | 2602 | 2667 | 2640 |
| KL_vp [18] | 2514 | 2530 | 2484 | 2615 | 2661 |
| KL_dual_vp [18] | 2626 | 2334 | 2256 | 2404 | 2433 |
| W_vp [18] | 2646 | 2417 | 2409 | 2474 | 2487 |
| MMD_pr [18] | 2583 | 2280 | 2285 | 2477 | 2435 |
| KL_pr [18] | 2241 | 2247 | 2181 | 2263 | 2233 |
| KL_dual_pr [18] | 2218 | 1984 | 2144 | 2215 | 2201 |
| W_pr [18] | 2241 | 2186 | 2284 | 2365 | 2344 |
| Best Baseline | 2839 | 2672 | 2602 | 2667 | 2661 |
| MOReL (Ours) | **3663 ±247** | **3305 ±413** | **3008 ±231** | **3329 ±270** | **3693 ±33** |

| Environment: Hopper-v2 | | | Partially trained policy: 1202 | | |
|---|---|---|---|---|---|
| Algorithm | Pure ($\mathcal{E}1$) | Eps-1 ($\mathcal{E}2$) | Eps-3 ($\mathcal{E}3$) | Gauss-1 ($\mathcal{E}4$) | Gauss-3 ($\mathcal{E}5$) |
| SAC [51] | 0.2655 | 661.7 | 701 | 311.2 | 592.6 |
| BC | 1330 | 129.4 | 828.3 | 221.1 | 284.6 |
| BCQ [15] | 1543 | 1652 | 1632 | 1599 | 1590 |
| BEAR [16] | 0 | 1620 | 2213 | 1825 | 1720 |
| MMD_vp [18] | 2291 | 2282 | 1892 | 2255 | 1458 |
| KL_vp [18] | 2774 | 2360 | 2892 | 1851 | 2066 |
| KL_dual_vp [18] | 1735 | 2121 | 2043 | 1770 | 1872 |
| W_vp [18] | 2292 | 2187 | 2178 | 1390 | 1739 |
| MMD_pr [18] | 2334 | 1688 | 1725 | 1666 | 2097 |
| KL_pr [18] | 2574 | 1925 | 2064 | 1688 | 1947 |
| KL_dual_pr [18] | 2053 | 1985 | 1719 | 1641 | 1551 |
| W_pr [18] | 2080 | 2089 | 2015 | 1635 | 2097 |
| Best Baseline | 2774 | 2360 | 2892 | 2255 | 2097 |
| MOReL (Ours) | **3642 ±54** | **3724 ±46** | **3535 ±91** | **3653 ±52** | **3648 ±148** |

| Environment: Walker-v2 | | | Partially trained policy: 1439 | | |
|---|---|---|---|---|---|
| Algorithm | Pure ($\mathcal{E}1$) | Eps-1 ($\mathcal{E}2$) | Eps-3 ($\mathcal{E}3$) | Gauss-1 ($\mathcal{E}4$) | Gauss-3 ($\mathcal{E}5$) |
| SAC [51] | 131.7 | 213.5 | 127.1 | 119.3 | 109.3 |
| BC | 1334 | 1092 | 1263 | 1199 | 1137 |
| BCQ [15] | 2095 | 1921 | 1953 | 2094 | 1734 |
| BEAR [16] | 2646 | 2695 | 2608 | 2539 | 2194 |
| MMD_vp [18] | 2694 | 3241 | **3255** | 2893 | **3368** |
| KL_vp [18] | 2907 | 3175 | 2942 | **3193** | 3261 |
| KL_dual_vp [18] | 2575 | **3490** | 3236 | 3103 | 3333 |
| W_vp [18] | 2635 | 2863 | 2758 | 2856 | 2862 |
| MMD_pr [18] | 2670 | 2957 | 2897 | 2759 | 3004 |
| KL_pr [18] | 2744 | 2990 | 2747 | 2837 | 2981 |
| KL_dual_pr [18] | 2682 | 3109 | 3080 | 2357 | 3155 |
| W_pr [18] | 2667 | 3140 | 2928 | 1804 | 2907 |
| Best Baseline | 2907 | 3490 | 3255 | 3193 | 3368 |
| MOReL (Ours) | **3709 ±159** | 2899 ±588 | **3186 ±92** | **4027 ±314** | 2828 ±589 |

| Environment: HalfCheetah-v2 | | | Partially trained policy: 4206 | | |
|---|---|---|---|---|---|
| Algorithm | Pure ($\mathcal{E}1$) | Eps-1 ($\mathcal{E}2$) | Eps-3 ($\mathcal{E}3$) | Gauss-1 ($\mathcal{E}4$) | Gauss-3 ($\mathcal{E}5$) |
| SAC [51] | 5093 | 6174 | 5978 | 6082 | 6090 |
| BC | 4465 | 3206 | 3751 | 4084 | 4033 |
| BCQ [15] | 5064 | 5693 | 5588 | 5614 | 5837 |
| BEAR [16] | 5325 | 5435 | 5149 | 5394 | 5329 |
| MMD_vp [18] | 6207 | **6307** | 6263 | **6323** | **6400** |
| KL_vp [18] | 6104 | 6212 | 6104 | 6219 | 6206 |
| KL_dual_vp [18] | **6209** | 6087 | **6359** | 5972 | 6340 |
| W_vp [18] | 5957 | 6014 | 6001 | 5939 | 6025 |
| MMD_pr [18] | 5936 | 6242 | 6166 | 6200 | 6294 |
| KL_pr [18] | 6032 | 6116 | 6035 | 5969 | 6219 |
| KL_dual_pr [18] | 5944 | 6183 | 6207 | 5789 | 6050 |
| W_pr [18] | 5897 | 5923 | 5970 | 5894 | 6031 |
| Best Baseline | 6209 | 6307 | 6263 | 6323 | 6400 |
| MOReL (Ours) | 6028 ±192 | 5861 ±152 | 5869 ±139 | 6026 ±74 | 5892 ±128 |

## C.6 Ablation Study with the Pure-partial dataset

Figure 6: Learning curve using naïve MBRL with the Pure-partial dataset. Contrast the learning exhibited by naïve MBRL in this figure with MOReL in Figure 4.

## C.7 Hyperparameter Guidelines and Ablations

We did not have resources to perform a thorough hyperparamter search, and largely used our intuitions to guide the choice of hyperparameters. We believe that better results are possible with hyperparameter optimization. First, we present the influence of the discrepancy threshold for differentiating known and unknown states. We first define the maximum discipancy in the dataset:

$$\text{disc}_{\mathcal{D}} = \max_{(s,a)\in\mathcal{D}} \max_{i,j} \|f_i(s,a) - f_j(s,a)\|$$

where $\mathcal{D}$ denotes offline dataset, and $f_i$ denotes $i^{th}$ dynamics model in the ensemble.

Table 6: Influence of discrepancy threshold on the Hopper-v2 task. We use a penalty of $0.0$ along with episode termination for visiting unknown regions in these experiments. We train all the cases for 1000 iterations, and report the average value over the last 100 iterations.

| Discrepancy Threshold | Value in P-MDP | Value in true MDP |
|---|---|---|
| $0.1 \times \text{disc}_{\mathcal{D}}$ | 1315.16 | 2082.21 |
| $0.2 \times \text{disc}_{\mathcal{D}}$ | 2479.92 | 3244.48 |
| $0.5 \times \text{disc}_{\mathcal{D}}$ | 3074.75 | 3359.66 |
| $1.0 \times \text{disc}_{\mathcal{D}}$ | 3543.23 | 3595.60 |
| $5.0 \times \text{disc}_{\mathcal{D}}$ | 3245.66 | 3027.59 |
| Naive-MBRL | 3656.08 | 2809.66 |

Our general observations and guidelines for hyperparameters are:

1. In Table 6, we first note that $0.1 \times \text{disc}_{\mathcal{D}}$ has the most amount of pessimism and Naive-MBRL has the least/no amount of pessimism. We observe that we obtain best results in the true MDP with an intermediate level of pessimism. Having either too much pessimism or no pessimism both lead to poor results, but for very different reasons that we outline below.

2. A high degree of pessimism makes policy optimization in the P-MDP difficult. The optimization process may be slow or highly noisy. This is due to non-smoothness introduced in the dynamics and reward due to abrupt changes involving early episode terminations. If difficulty in policy optimization is observed in the P-MDP, we recommend considering reducing the degree of pessimism.

3. With a lack or low degree of pessimism, policy optimization is typically easier, but the performance in the true MDP might degrade. If it is observed that the value in the P-MDP overestimates the value in the true MDP substantially, then we recommend increasing the degree of pessimism.

4. For the tasks considered in this work, positive rewards indicate progress towards the goal. Most of the locomotion tasks involve forward velocity as the primary component of the the reward term. In these cases, we observed that the choice of reward penalty for going into unknown regions did not play a crucial role, as long as it was $\leq 0$. The degree of influence of this parameter in other environments is yet to be determined, and beyond the scope of our empirical study.