[Reviews · NeurIPS 2020]

Review 1

Summary and Contributions: This paper presents an interesting approach to offline MBRL using pessimistic MDP. The authors provide an approximate lower bound of performance in the environment by the policies learned in the P-MDP. Some insights about the claims and approaches for learning the P-MDP are provided through numerical experiments. ------- Based on the author's response and overall evaluation, I recommend that the paper be accepted.

Strengths: I think the main strength of the paper is the theoretical results provided by the authors in section 4.1. I think the result of lower bounding the performance in the environment with the P-MDP learned by the authors is novel and interesting. I also like that the authors were able to illustrate the transfer from P-MDP to the environment.

Weaknesses: The idea of offline model learning could be related to system id approaches in control theory. However, this still remains difficult for high dimensional system. The authors don't provide a novelty in fixing this issue. However, they use an ensemble of parameterize models to quantify uncertainty and then, use, I would say a rather, ad-hoc way to define their USAD using a hyperparameter. This again makes the algorithm susceptible to instability based on the choice of this hyper parameter. It would have been nice if the authors would have explored the effect of model learning on the performance of the controller. Would it be possible to demonstrate the affect of this hyperparameter on the policy? While the authors provide an intuition that the policy for data logging does affect their final policy, is there an intuition about collecting good data from a system? Consider that now we interact with a new system-- how can we log data to implement this method on the system? When using a pure-random policy, the performance of the controller seems to drop quite a lot in the proposed method. It is not clear to me how much the policy performance depends on the choice of the method used to compute the policy? Why was the method NPG chosen for policy computation? Did the authors explore any other method for policy computation?

Correctness: I like the theoretical results that the authors have provided in section 4.1, and they show that for the examples demonstrated in section 5 that the P-MDP provides a lower bound to the performance in the environment. I think this might be the strongest part of the paper.

Clarity: The paper is very well written. The theoretical results are well described as well as the results are well explained.

Relation to Prior Work: yes, the authors have pointed out relevant differences from related work in this field.

Reproducibility: Yes

Additional Feedback:


Review 2

Summary and Contributions: The paper proposes a model-based offline RL algorithm based on tracking the uncertainty in the learned dynamics model and making uncertain states transition to a negative reward absorbing state. It shows some theoretical analysis of performance and good results on mujoco-based offline RL benchmarks.

Strengths: The paper is well written with a good description of related work and a good empirical comparison with positive results.

Weaknesses: My enthusiasm for the paper is limited by the fact that these kinds of ideas have already been well explored in the historical model-based RL literature. The real contribution of the paper is mostly to carry the ideas forward to modern offline RL setups and benchmarks. On the plus side, the empirical results are good. Theorem 1 is ok but doesn't provide much novel insight. Basically if a policy doesn't drive toward uncertain states very quickly, its value in the original and the pessimistic MDP is similar, which is as expected. Analyzing this via hitting times is interesting.

Correctness: The empirical results show standard deviations over 5 random seeds which is a nice improvement over previous work that did not report theirs. However, the appropriate thing to report would be standard errors. In particular, if your intent is to indicate results where you do not have the best average performance but are within the uncertainty of the evaluation (as you do) then standard error should be used.

Clarity: In proposition 4, why is gamma constrained [0.95,1)? I'm assuming 0.95 is not important, you just need some specific value for it not too close to zero?

Relation to Prior Work: Yes

Reproducibility: Yes

Additional Feedback: I might have missed it, but I believe you didn't define D_TV before using it in equation 2.


Review 3

Summary and Contributions: This paper presents a model-based offline reinforcement learning algorithm, MOReL. MOReL constructs a pessimistic MDP (P-MDP) using an unknown state-action detector, which largely penalizes the 'unknown' state-action regions. Then, the policy is optimized via the constructed P-MDP. Theoretically, it is proven that MOReL is nearly minimax optimal. Empirically, MOReL is competitive or outperforms the state-of-the-art offline RL algorithms.

Strengths: Offline reinforcement learning is an important problem, which can enable RL to be applied to many real-world problems where the data-collection cost or safety is crucial. This work provides the first (deep) model-based offline RL algorithm that has a theoretical guarantee as well as strong empirical performance, which are novel contributions of the work.

Weaknesses: I didn't find a significant weakness of the paper.

Correctness: I didn't read the full proof, but the statements look sound and correct. The proposed algorithm is sound and well-backed by the theorem.

Clarity: The paper is well-written and easy fo follow.

Relation to Prior Work: Overall, the contributions of the work are clearly discussed, compared to other offline RL works.

Reproducibility: Yes

Additional Feedback: Most of recent offline RL algorithms rely on policy regularization where the optimizing policy is prevented from deviating too much from the data-logging policy. Differently, MOReL does not directly rely on the data-logging policy but exploits pessimism to a model-based approach, providing another good direction for offline RL. Overall, I lean toward the acceptance of the paper. - MOReL constructs a pessimistic MDP, where all unknown (state,action) pairs are penalized equally. However, it would be more natural to penalize more to more uncertain states. For example, one classical model-based RL algorithm (MBIE-EB) constructs an optimistic MDP that rewarding the uncertain regions by the bonus proportional to the 1/sqrt(N(s,a)) where N(s,a) is the visitation count. In contrast, but similarly to MBIE-EB, we may consider a pessimistic MDP that penalizes the uncertain regions by the penalty proportional to the 1/sqrt(N(s,a)). - It seems that theorem 1 always encourages to use the smallest alpha, i.e. alpha=0. However, when alpha=0, USAD of Eq. (2) will always output TRUE, where MOReL would not work properly. How is it justified to use alpha greater than zero for USAD? - It would be great to see how sensitive the performance of the algorithm with respect to kappa in the reward penalty and threshold in USAD. - To claim SOTA performance, MOReL may have to be compared to more recent works on offline RL such as [1] and [2]. [1] Siegel et al., Keep Doing What Worked: Behavioral Modelling Priors for Offline Reinforcement Learning, 2020 [2] Lee et al. Batch Reinforcement Learning with Hyperparameter Gradients, 2020 == post rebuttal I have read the author response, and I still remain on the acceptance of the paper.

[Author Response · NeurIPS 2020]

We thank all the reviewers for their thoughtful feedback. We highlight that all the reviews were positive with a few specific questions, which we hope to address in our response below.

**Reviewer # 2**

1. Hyper-parameters : We will include an ablation study in the final version of paper.
2. How to collect "good" data for a new task : We first wish to clarify that in offline RL, the primary motivation is to use existing offline datasets. As a consequence, we typically treat the dataset as fixed and given to the agent; as opposed to the agent (or researcher) having the luxury of choosing the dataset. Having said that, an ideal dataset for offline RL would be one that not only has high support overlap with the optimal policy, but also one that enables a large hitting time, as suggested by our proposition and lower bound. Designing exploratory policies for purposes of data collection is an exciting direction for future work but outside the scope of current submission.
3. Choice of NPG : We used NPG for its conceptual and implementation simplicity. A number of prior papers have successfully used NPG and shown impressive results. Furthermore, our MOReL framework is modular and has a clear separation between learning the P-MDP and optimizing a policy in the P-MDP. We conjecture that most algorithms (e.g. PPO, SAC etc) for optimizing the policy in the P-MDP would yield similar results.

**Reviewer # 5**

1. Contributions of our work : To our knowledge, our work is the first to study a model-based approach to offline RL, apart from Ross et al. which provided negative results for a naive algorithm. While there has been extensive work on model-based RL and offline RL individually, their intersection has been explored only sparsely. As most reviewers concurred, offline RL is an important learning paradigm that can expand the applicability of RL. We develop a new framework for offline RL that utilizes learned models and show that it is mini-max optimal. We also demonstrate state of the art experimental results.
   Our survey of related work is extensive with 86 citations (kindly also see expanded related works in appendix). We are also happy to cite and discuss any additional related work that the reviewers may point out.
2. Theoretical insights : While it is intuitively clear that if a policy does not drive too quickly towards unknown states, Theorem 1 presents a *precise, quantitative* bound using *hitting times*. Corollary 3 further bounds this in terms of mismatch in the support of state-action visitation distributions. In contrast, prior works only consider settings where there is no support mismatch. Proposition 4 shows that the bound in Corollary 3 is best possible up to logarithmic factors, demonstrating minimax optimality of MOReL.
3. Use standard errors in table : Thank you for the suggestion. We followed common practice to report standard deviations, but we are happy to report standard errors if it is more appropriate in the reviewer's opinion. Note however that our claim of SOTA results in 12 out of 20 environments is based on our average scores, which remains unaffected by choice of error bars. We also highlight that prior work does not report any error bars, and also tune hyper-parameters on a seed-specific basis. In contrast, we use the same hyper-parameters across all seeds.
4. Proposition 4 : The value of 0.95 comes from requiring $\gamma$ to satisfy certain inequalities in Lines 579 and 581 in Appendix A. Since our goal is to show that the $(1-\gamma)^{-2}$ dependence in Corollary 3 is optimal, it is okay to assume that $\gamma \in [0.95, 1]$ (since, if $\gamma < 0.95$, then $(1-\gamma)^{-2}$ is bounded by a constant).

**Reviewer # 6**

1. Alternate ways to penalize uncertain states : Our particular approach to penalizing unknown states enables detailed theoretical analysis while also demonstrating SOTA experimental results on well studied domains that require function approximation. It would make for an interesting future work to study if similar results (theoretical and/or experimental) can be obtained with alternate approaches, but is outside the scope of our submission.
2. Choice of $\alpha$ : Note that there are two competing terms in the sub-optimality bound (Corollary 1). Decreasing $\alpha$ decreases $(1-\gamma)^{-2} \cdot (4\gamma R_{\max}) \cdot \alpha$, but also has the effect of decreasing the number of "known" states. This in-turn reduces the hitting time, and increases $(1-\gamma)^{-1} \cdot 2R_{\max} \cdot \mathbb{E}\left[\gamma^{T_{\mathcal{U}}^{\pi^*}}\right]$ in the bound. Thus, an appropriate choice of $\alpha$ that balances the two terms is required, and can be treated as a hyper-parameter.
3. Hyper-parameters : We will include an ablation study of hyper-parameters in the final version of the paper.
4. Comparison to more recent work : Thank you for the pointers to these very interesting papers! First, we wish to highlight that these are *very recent* papers making comparisons with them difficult – especially at the time of NeurIPS submission. Furthermore, these papers report results on non-standard domains compared to most prior work. For example, ABM uses DeepMind control suite while BOPAH uses a different data logging policy. This makes a direct comparison with published results impossible. In this submission, we used identical setups to most prior papers for a fair and transparent comparison.

[Meta-Review · NeurIPS 2020]

All three reviewers have favourable opinion towards this paper. There are some minor questions or comments, but they can be addressed without requiring another round of reviewing. Therefore, I recommend acceptance of this work. I encourage the authors to incorporate the reviewers' comments and concerns as much as possible.